A population genetic assessment of coral recovery on highly disturbed reefs of the Keppel Island archipelago in the southern Great Barrier Reef

van Oppen Madeleine J.H. 1 2 3 m.vanoppen@aims.gov.au
Lukoschek Vimoksalehi 3
Berkelmans Ray 1
Peplow Lesa M. 1
Jones Alison M. 4
1 Australian Institute of Marine Science , Queensland , Australia
2 School of BioSciences, The University of Melbourne , Parkville, Melbourne, Victoria , Australia
3 ARC Centre of Excellence for Coral Reef Studies, James Cook University , Townsville, Queensland , Australia
4 Central Queensland University , Rockhampton, Queensland , Australia
Medina Mónica
Electronic publication date: 2015 Jul 23
Publication date: 2015
Volume: 3
Electronic Location ID: e1092
Received 2015 May 29; Accepted 2015 Jun 18
Copyright: © 2015 van Oppen et al.
Copyright year: 2015
Copyright holder: van Oppen et al.
License: This is an open access article distributed under the terms of the Creative Commons Attribution License, which permits unrestricted use, distribution, reproduction and adaptation in any medium and for any purpose provided that it is properly attributed. For attribution, the original author(s), title, publication source (PeerJ) and either DOI or URL of the article must be cited.
License URL: https://creativecommons.org/licenses/by/4.0/

Keywords: Acropora millepora, Microsatellites, Gene flow, Population structure, Genetic diversity, Coral reef management

Funding: Central Queensland University Australian Institute of Marine Science This research was supported by a Central Queensland University Research Advanced Award Scheme Postdoctoral Fellowship to AM Jones, and the Australian Institute of Marine Science. The funders had no role in study design, data collection and analysis, decision to publish, or preparation of the manuscript.

==============================
Coral reefs surrounding the islands lying close to the coast are unique to the Great Barrier Reef (GBR) in that they are frequently exposed to disturbance events including floods caused by cyclonic rainfall, strong winds and occasional periods of prolonged above-average temperatures during summer. In one such group of islands in the southern GBR, the Keppel Island archipelago, climate-driven disturbances frequently result in major coral mortality. Whilst these island reefs have clearly survived such dramatic disturbances in the past, the consequences of extreme mortality events may include the loss of genetic diversity, and hence adaptive potential, and a reduction in fitness due to inbreeding, especially if new recruitment from external sources is limited. Here we examined the level of isolation of the Keppel Island group as well as patterns of gene flow within the Keppel Islands using 10 microsatellite markers in nine populations of the coral, Acropora millepora. Bayesian cluster analysis and assignment tests indicated gene flow is restricted, but not absent, between the outer and inner Keppel Island groups, and that extensive gene flow exists within each of these island groups. Comparison of the Keppel Island data with results from a previous GBR-wide study that included a single Keppel Island population, confirmed that A. millepora in the Keppel Islands is genetically distinct from populations elsewhere on the GBR, with exception of the nearby inshore High Peak Reef just north of the Keppel Islands. We compared patterns of genetic diversity in the Keppel Island populations with those from other GBR populations and found them to be slightly, but significantly lower, consistent with the archipelago being geographically isolated, but there was no evidence for recent bottlenecks or deviation from mutation-drift equilibrium. A high incidence of private alleles in the Keppel Islands, particularly in the outer islands, supports their relative isolation and contributes to the conservation value of the archipelago. The lack of evidence for genetic erosion, in combination with our observation that the North Keppel Island population samples collected in 2002 and 2008, respectively, exhibited a pairwise genetic distance of zero, supports previous published work indicating that, following bleaching, Acropora corals in the Keppel Islands predominantly recover from regrowth of small amounts of remaining live tissue in apparently dead coral colonies. This is likely supplemented by recruitment of larvae from genetically similar, less disturbed populations at nearby reefs, particularly following extreme flood events.

Introduction

Coral reefs along the East Australian coastline are shaped by a range of factors and forces that include coastal geomorphology, freshwater inundation and sediment runoff, hydrodynamics, unusually warm summer sea surface temperatures caused by climate warming, as well as local weather patterns. These forces cause recurring perturbations and in some regions result in frequent, high levels of coral mortality. One of the largest inshore reef systems of the southern Great Barrier Reef (GBR) is comprised of the fringing reefs surrounding the 15 islands of Keppel Bay, located ∼12 km from the mainland coast. The Keppel Islands are renowned for their high disturbance regime, causing repeated widespread coral mortality. A major flooding event occurred here in 1991 (Byron & O’Neill, 1992; Furnas, 2003; Jones & Berkelmans, 2014), which caused bleaching and a mortality of almost 85% of all corals and total mortality of Acropora spp. down to 1.3 m below lowest tide level (van Woesik, DeVantier & Glazebrook, 1995). Thermal mass coral bleaching affected >60% of the corals in this area in 1998, 2002 and 2006 and caused significant coral cover loss (e.g., ∼40% loss in 2006; Jones, Berkelmans & Houston, 2011), particularly in shallow (0–6 m) reef areas (Berkelmans et al., 2004; Jones et al., 2008; Diaz-Pulido et al., 2009).

Typically, larval recruitment on tropical reefs occurs either from local, sexually mature and healthy corals or from nearby and occasionally distant source populations. Spatial and temporal patterns of recruitment are often variable and can be driven by factors such as local wind patterns, prevailing winds, the direction and strength of wind-driven currents, the proximity of other reefs, water depth, and structural complexity (Hughes et al., 2000; Whitaker, 2004; Underwood et al., 2007; van Oppen et al., 2008; Almany et al., 2009). Preliminary genetic analyses indicate the Keppel Islands are likely an isolated system (van Oppen et al., 2011). Because larval input from external sources is generally considered crucial for recovery on reefs that have suffered extensive coral mortality (Lukoschek et al., 2013), it is important to validate that larval dispersal into the Keppel Island archipelago is restricted. However, the importance of external larval sources may be overestimated if partial, rather than whole colony, mortality is common and rapid regrowth of surviving tissues ensues (Riegl & Piller, 2001; Gilmour et al., 2013), a process that is a key mechanism of recovery from bleaching for Acropora spp. in the Keppel Islands (Diaz-Pulido et al., 2009).

Here we examine the mechanisms underlying recovery in the common reef-building coral, Acropora millepora (Cnidaria; Scleractinia; Acroporidae), in the Keppel Islands using a population genetics approach. Specifically, we explore genetic structure, connectivity and diversity on nine shallow reefs throughout the Keppel Island region using high-resolution DNA microsatellite markers. We also compare population genetic diversity and local population genetic structure of A. millepora in the Keppel Islands to that of 19 reefs spanning much of the latitudinal range of the GBR and including one of the nine Keppel Island reefs sampled six years earlier (van Oppen et al., 2011). We discuss the implications of our findings in terms of the future management of the Keppel Island reefs.

Material and Methods

The Keppel Bay Island archipelago lies ∼30 km north of the mouth of the Fitzroy River near Rockhampton (Fig. 1). Like much of the inshore GBR, the reefs principally fringe the bay heads of the islands and, to a lesser extent, the rocky coastal headlands (Hopley, 2006). Wherever substantial reefs exist, these are dominated by large stands of fast-growing ‘structural’ species such as the acroporids, pocilloporids and poritids (Jones, Berkelmans & Houston, 2011). One such coral, A. millepora (Scleractinia: Acroporidae), grows prolifically between 0–6.0 m (lowest astronomical tide) forming shallow, expansive reef flats on the leeward shores of islands in the Bay (Fig. 2). A. millepora is a common and ecologically important species on the GBR, particularly on the inshore reefs. Like most Acroporidae, A. millepora reproduces sexually via a single annual broadcast spawning event, and to a lesser extent via asexual reproduction through fragmentation (Smith & Hughes, 1999).

Figure 1 Maps of sampling locations of Acropora millepora (black circles) from: (A) van Oppen et al. (2011) and (B) this study.

Colour plots (C) to (I) are TESS results using the admixture model and K = 5, in which each bar represents an individual coral colony and the five colours represent the five genetic clusters. Plots (C) to (H) correspond to reefs sampled in boxes (C) to (H) on map (A), while plots (J) to (I) correspond to reefs sampled in map (B).

Figure 2 Image showing a typical shallow water reef in the Keppel Island archipelago, dominated by Acropora millepora.

Photo credit: Alison Jones.

Coral Sampling

Branches of A. millepora were collected between December 2008 and April 2009 under the Great Barrier Reef Marine Park Authority collection permit numbers G09/30237.1 and G08/26114.1, and their genotypes were determined at 10 microsatellite loci. The nine sampled reefs (Halfway Island, Outer Rocks, Man and Wife Rocks, Barren Island, North Keppel Island, Passage Rocks, Miall Island, Halftide Rocks and Humpy Island; Fig. 1) were chosen to include both inshore and offshore islands within the archipelago.

At each site, 29–50 samples were collected from colonies located at depths between 0–6.0 m by removing a single branch from each colony. Samples were preserved in absolute ethanol. Samples at each site were collected from areas less than 500 m2, targeting colonies >5 m apart on haphazard swim trajectories using SCUBA. This approach minimises the likelihood of sampling colonies generated asexually via fragmentation from the source colony, as fragments of A. millepora on reef flat habitats are rarely dispersed further than 4 m from their source colonies and typically have low survival and reattachment rates (Smith & Hughes, 1999).

Genetic Characterisation

DNA was extracted from the preserved samples based on a slightly modified version of the method by Wilson et al. (2002). PCR primers and protocols for the ten microsatellite loci are described in van Oppen et al. (2011) and Wang, Zhang & Matz (2009). Twelve microsatellite markers were used in the PCR reactions and run in four multiplex reactions (Table S1); however, two loci were not used because of inconsistent amplification success.

Data Analysis

MegaBACE Genetic Profiler Software Suite version 2 (GE Healthcare, Little Chalfont, UK) was used to determine the fragment sizes (alleles) of all samples. All automatic scoring was checked manually, and samples that yielded ambiguous or no signal were re-amplified and re-run or removed from the analysis. The new data acquired in this study were first analysed separately and subsequently combined with previously obtained data on the same species and using the same loci, but from 20 GBR locations spanning 12° of latitude (van Oppen et al., 2011) and including one site in the Keppel Islands (Nth Keppel Island). Because the van Oppen et al. (2011) data were scored with a different software package (CEQ8800 system software, version 10), the possibility existed that alleles were scored differently and a shift in allele size had occurred between the two methods. A subset of 3–5 samples from the van Oppen et al. (2011) data set harbouring the most common alleles was therefore selected for each locus, and rescored using the MegaBACE software. Based on this comparison, the allele sizes of all samples and loci were adjusted to match the van Oppen et al. (2011) study. The combined data set of the two studies is available in Data S1.

The probabilities of identity by random sexual mating (Waits, Luikart & Taberlet, 2001) were calculated using an AMOVA (Analysis of Molecular Variance) approach (Excoffier, Smouse & Quattro, 1992) in GenAlEx v6.501 (Peakall & Smouse, 2006). Individuals sharing the same multilocus genotype (MLG) were inferred to be clone mates if probabilities of identity by random sexual mating were small. If asexual reproduction was inferred, all but one individual with this MLG were removed prior to further data analysis.

Genotypic Linkage Disequilibrium (LD) was assessed in GENEPOP (web version 4.0.10) by estimation of exact p-values using the Markov chain method (Raymond & Rousset, 1995) using default settings. A previous study for A. millepora using the same loci that included one site (Nth Keppel) from the Keppel Islands, showed that despite the presence of null alleles, heterozygote deficits were mostly due to biological rather than methodological factors (van Oppen et al., 2011). Despite the occurrence of some instances of deviations from HWE (Table S2), all analyses were therefore conducted on data uncorrected for null alleles.

Genetic diversity, population structure, gene flow and isolation by distance

Various aspects of genetic diversity and uniqueness were estimated in GenAlEx v6.501 including the number of alleles per locus (Na), allelic richness (Ar), allelic evenness (Ae), observed (HO) and expected (HE) heterozygosities and private alleles. Differences in rarefacted allelic richness (using 22 individuals per site, the smallest sample size in the data set) between the nine Keppel Island sites and the 19 other sites from throughout the GBR were assessed in FSTAT 2.9.3 using a Mann–Whitney U test (Goudet, 1995).

Populations that have experienced a recent reduction in their effective population size exhibit a reduction in the allele numbers and transient heterozygous (HO) excess at polymorphic loci compared to that under HWE (HE) (Cornuet & Luikart, 1996). If HWE is assumed (i.e., no recent bottleneck), there is an equal probability of having a positive or a negative difference between the observed and the expected heterozygosities. In contrast, following a recent bottleneck, heterozygous excess is expected to occur more often than heterozygous deficit. Therefore, if the number of loci for which there is heterozygous excess is significantly larger than that for which there is a heterozygous deficit, a recent bottleneck can be inferred (Luikart & Cornuet, 1998). The heterozygosity distribution under the assumption of HWE and the infinite allele mutation model was calculated for each of the nine Keppel Island sites and for each locus in the software package Bottleneck 1.2.02. Bottlenecks are also expected to change the allele frequency distribution (Cornuet & Luikart, 1996). Therefore, the allele frequency distribution was established to see whether it was approximately L-shaped (as expected under HWE) or not.

Population structure within the Keppel Islands and the combined data sets was estimated using FST values calculated using an AMOVA approach (Excoffier, Smouse & Quattro, 1992) in GenAlEx v6.501 (Peakall & Smouse, 2006) with significance tested using 999 permutations. Genetic differentiation between sites was estimated in the following ways: (1) FST values were calculated using an AMOVA approach in GenAlEx v6.501. To assess the significance of differentiation between sites, we applied a Fisher exact test (Goudet, 1995) using Genepop v4.0 with the default Markov chain parameters. Statistical significance for all pairwise tests was adjusted for multiple comparisons by the B-Y False Discovery Rate (FDR) method (Narum, 2006). (2) Jost’s (2008) actual measure of differentiation (Dest) was computed in SMOGD version1.2.5 (Crawford, 2010). To visualise the genetic relationships among populations, the genetic distance measures between pairs of Keppel Island sites were plotted using a Principal Coordinates Analysis (PCoA) with GenAlEx v6.501. To determine whether there was a pattern of isolation-by-distance (IBD), pairwise Dest values were regressed onto over-water distances between sites and significance tested using Mantel permutation test in IBD Web Service (Jensen, Bohonak & Kelley, 2005).

Two fully Bayesian model-based clustering methods implemented in the programs STRUCTURE ver. 2.3.3 (Pritchard, Stephens & Donnelly, 2000) and TESS ver. 2.3 (Chen et al., 2007; François & Durand, 2010) were used to further examine spatial genetic structure for the Keppel Islands (n = 370) and Keppel Islands plus the GBR (n = 1,292) datasets. STRUCTURE analyses were conducted using both the admixture and no-admixture models, each with correlated allele frequencies, using the sampling sites as prior (LOCPRIOR), which has been shown to better resolve genetic structure when there is low genetic divergence (Hubisz et al., 2009). MCMC chains used a burn-in of 50,000 chains followed by 500,000 of MCMC replications. Ten independent chains were run for each K from K of 1 to 9 for the KI data and K of 1 to 15 for the combined data. In each case, the most likely value of K was evaluated using the method of Evanno, Regnaut & Goudet (2005) as implemented in STRUCTURE HARVESTER (Earl, 2009). STRUCTURE implements an algorithm that puts a strong emphasis on the prior of the existence of clusters, which may make it prone to errors when geographical sampling is discrete along clines (Chen et al., 2007). TESS aims to address this issue by using a spatially continuous prior based on the geographical coordinates of each sampled individual. TESS was run using the CAR admixture model, which assumes spatial autocorrelation of the genomes of individuals in closer geographical proximity compared with those further apart. The strength of this autocorrelation is represented by a spatial interaction parameter (ψ), which was set to the default value of 0.6 for analysis. TESS was run with a burn-in of 10,000 sweeps followed by 25,000 sweeps, with 20 independent runs conducted for each value of K from K of 2 to 9 for the KI data and K of 2 to 15 for the combined KI plus GBR data (TESS does not implement analyses for K = 1). For each value of K, the ten runs with the lowest DIC scores were used to calculate the average DIC and evaluate the most likely number of genetic clusters. The coefficient of ancestry was calculated for each individual across all runs for the most likely value of K in CLUMPP version 1.1.2 (Jakobsson & Rosenberg, 2007) and results visualized with the program DISTRUCT version 1.1 (Rosenberg, 2004).

GeneClass2 (Piry et al., 2004) was used to examine first generation migrants (i.e., recent gene flow) within the Keppel Island archipelago (only the Keppel Island data were used for this analysis). In the first step of this analysis, migrants were identified using the criteria and computational algorithm of Rannala & Mountain (1997) with 10,000 simulated genotypes and an alpha of 0.01. The test statistic Lh was used as not all potential source populations had been sampled (Paetkau et al., 2004). Migrants were excluded from the data set, and this adjusted data set served as the reference data set to which migrants were assigned. Migrants were assigned to populations if the assignment probabilities were greater than 0.1.

Results

Genetic diversity

All loci were polymorphic in all populations sampled, with numbers of alleles ranging from 2 to 17 (Table S2). Expected heterozygosities ranged from 0.232 to 0.885 (Table S2). Three MLGs in the Keppel Islands data set were repeated twice each; two of these MLGs occurred at Barren Island and one at Man & Wife Rocks. One sample from each pair was removed prior to further analyses. The resulting data set consisted of 370 MLGs from nine locations. Five, three, three, six, one, one, and two instances (out of 45 pairwise comparisons within each population) of LD were observed in Barren Island, Outer Rocks, Man & Wife Rocks, Halftide Rocks, Nth Keppel Island, Humpy Island, and Passage Rocks, respectively (Table S3). No cases of LD were observed in the Miall and Halfway Island populations.

In the combined data set, rarefacted allelic richness was slightly, but statistically significantly lower between the Keppel Island populations and all other GBR populations included in this study (6.7 vs. 7.3 alleles respectively, p = 0.006). Plots of allelic evenness (Figs. S1A and S1B) confirm that, with the exception of Man & Wife Rocks, genetic diversity is consistently lower in the Keppel Islands compared to elsewhere on the GBR. The Bottleneck analyses indicated all loci in all populations fit the mutation-drift equilibrium, and there were no deviations from an L-shaped allele frequency distribution, suggesting no recent bottlenecks have occurred.

Private alleles were found in 54 out of 1,292 colonies of A. millepora from the combined GBR-Keppel Islands data set, 23 (43%) of which occurred in the Keppel Islands. Given the relatively small sample size from Keppel Island populations (320 out of 1,292, i.e., 25% of the total sample size), private alleles are overrepresented in this archipelago. Twenty-one of the 23 Keppel Island samples with private alleles were from the outer island group.

Population structure, gene flow and isolation by distance

AMOVA showed that 5% of the total variance in the Keppel Island data set was partitioned among populations (Global FST = 0.055, p < 0.001). Pairwise FST values were significant for all comparisons (B-Y FDR; αCRIT = 0.012) except for Halfway Island-Miall Island, Halfway Island-Nth Keppel Island and Miall Island-Nth Keppel Island (Table S4). The Barren Island population was highly divergent, with most FST values >0.1. Twenty-six of 36 pairwise Dest values were statistically significant, and the Dest values also indicated that the Barren Island population was highly divergent, with most values >0.2 (Table 1). This pattern is clearly visualised in the PCoA of pairwise Dest values (Fig. S2). There was no evidence of IBD (r2 = 0.07, p = 0.150; Fig. S3), which is consistent with the pattern of some geographically disparate pairs of sites being genetically similar (e.g., Halfway Island vs. Nth Keppel Island; Humpy Island vs. Nth Keppel Island; Table 1), while other geographically proximate sites are genetically divergent (e.g., Man & Wife Rocks vs. Halftide Rocks; Table 1).

Table 1 Pairwise Dest values, below diagonal, p-values above diagonal.

	Barren Island	Halftide Rocks	Halfway Island	Humpy Island	Man & Wife Rocks	Miall Island	Nth Keppel Island	Outer Rocks	Passage Rocks	
Barren		0.001	0.001	0.001	0.001	0.001	0.001	0.001	0.001	
Halftide	0.253		0.027	0.001	0.001	0.001	0.001	0.001	0.001	
Halfway	0.212	0.012		0.024	0.001	0.017	0.392	0.088	0.001	
Humpy	0.192	0.038	0.015		0.001	0.080	0.020	0.016	0.003	
Man & Wife	0.176	0.077	0.054	0.059		0.001	0.002	0.009	0.001	
Miall	0.239	0.042	0.017	0.010	0.084		0.070	0.037	0.001	
Nth Keppel	0.221	0.030	0.001	0.018	0.064	0.011		0.097	0.001	
Outer	0.175	0.040	0.009	0.020	0.041	0.014	0.010		0.001	
Passage	0.242	0.119	0.091	0.032	0.125	0.100	0.089	0.110		
Notes.

Most values are statistically significant; non-significant values have shaded background, and p-values larger than adjusted α are printed in bold face (adjusted α = 0.012).

Forty-two of the 370 Keppel Island individuals included in this study were identified as first generation migrants based on the GeneClass2 analysis (Table S5). In the outer Keppel Islands, five out of 29 (Barren), three out of 28 (Man & Wife) and six out of 50 (Outer) were identified as recent migrants. Four of these could not be assigned (i.e., had assignment probabilities <0.1 to all sampled populations), seven had the greatest probabilities for assignment to other outer reefs, three had high assignment probabilities to both inner and outer Keppel Island populations, while none were assigned to inner island populations only. Of the 28 migrants identified in the inner islands, 19 were assigned. Eight of these were assigned to one or more of the outer island populations, six to both inner and outer populations, and five to other inner island populations. These results suggest recent gene flow has occurred both within and between island groups, and that gene flow occurs from east to west and vice versa, but likely more frequently from east (outer islands) to west (inner islands).

STRUCTURE using the admixture model indicated that two or three genetic clusters best explained the genetic patterns of the multilocus genotypes of the 370 colonies of A. millepora in the Keppel Islands, with highest ΔK for K = 2 followed by K = 3 (Fig. S4). Similarly, TESS DIC scores declined sharply between K = 2 and K = 3 and then declined much more slowly while variances in DIC increased markedly, providing support for three genetic clusters (Fig. S4). STRUCTURE using the no-admixture model did not provide a clear result. TESS and STRUCTURE using the admixture model for K = 2 returned almost identical genetic patterns, with Passage Rocks, Halftide Rocks, Halfway, Humpy, Miall and Nth Keppel Islands forming a panmictic cluster, while Barren Island, Man & Wife and Outer Rocks had some individuals from the panmictic cluster and others from the second genetic cluster (Fig. S5). TESS for K = 3 returned a similar pattern to K = 2 except that Man & Wife Rocks was distinct from Barren Island and Outer Rocks, with colonies that did not belong to the panmictic cluster belonging to the third genetic cluster (Fig. S5). By contrast, STRUCTURE for K = 3 found admixture between the panmictic and the third genetic cluster within all individuals at Passage Rocks and approximately half the individuals at Humpy Island (Fig. S5). This result, combined with the higher ΔK for K = 2 than K = 3 suggests that, unlike TESS, the algorithm implemented in STRUCTURE was unable to resolve Man & Wife Rocks as a distinct genetic cluster.

For the combined GBR plus Keppel Island data set, STRUCTURE results showed that ΔK was highest for K = 2 followed by K = 3 and then peaked again at K = 5, while TESS DIC values declined steeply between K = 2 and K = 5 and then declined more slowly (Fig. S6). Although ΔK for K = 5 (ΔK = 20) was smaller than for K = 2 (ΔK = 230) and K = 3 (ΔK = 35), all were much larger than for all other values of K (typically ΔK < 1). Given that TESS clearly delineated three genetic clusters for the Keppel Islands alone, we present K = 5 for the combined dataset. All sites in the Keppel Island archipelago were genetically distinct from GBR populations in the far northern, northern and central GBR reefs, as well as most southern GBR reefs except High Peak (Fig. 1), which may receive larvae from the Keppel Islands via the predominantly north-east flowing sea surface currents in this part of the GBR (Luick et al., 2007). In particular, Barren Island and some individuals from Man & Wife and Outer Rocks belonged to a genetic cluster not found elsewhere on the GBR. Temporal samples from Nth Keppel Island (July 2002, van Oppen et al., 2011; van Oppen et al., 2008, this study) were genetically similar (Fig. 1) and had FST values not significantly different from zero (results not shown).

Discussion

Limited gene flow between inner and outer island clusters

The Barren Island population is a genetic outlier with Dest values ranging from 0.175 to 0.253 (Table 1), and most of the individuals sampled belong to a genetic cluster distinct from any other cluster observed on the GBR (Fig. 1). The reasons underlying the extreme genetic distinctiveness of this population are unclear. Outer and Man & Wife Rocks have smaller numbers of individuals of the same distinct genetic affinity. Despite this, all three outer island populations contain some individuals that are of the inner islands genetic affinity. Further, they show a signature of admixture with some colonies being comprised of the distinct as well as the more typical inner island genetic cluster, suggesting some level of gene flow exists between outer and inner islands. This was confirmed by assignment tests, which in addition suggested gene flow is higher from east to west than from west to east, consistent with the predominant direction of sea surface currents (Luick et al., 2007). A genetic parentage study of two coral reef fish species found that recent dispersal rates were higher among the inner Keppel Islands than between Barren Island and the inner islands (Harrison et al., 2012), consistent with our observations for A. millepora.

Coral larval competency is unlikely a limiting factor for gene flow of A. millepora as larvae of this species are competent to metamorphose and settle around 4–5 days after spawning (Babcock & Heyward, 1986), with maximum rates of metamorphosis occur at eight days after spawning (Heyward & Negri, 1999). Maximum longevity of Acropora coral larvae in the water column, however, is much longer (∼60–200 days) (Nishikawa, Katoh & Sakai, 2003; Graham, Baird & Connolly, 2008). Larval dispersal is affected by surface water circulation patterns. Numerical particle experiments indicate that during the northward-current season (the austral summer in which coral mass spawning takes place), cross-shelf particle dispersal is limited (Luick et al., 2007), likely contributing to the population structure observed here. Alternatively, realised dispersal may be lower than the actual dispersal potential due to maladaptation of outer island genotypes to inner island environmental conditions and vice versa (Prada & Hellberg, 2014). While the environmental factors light, temperature and habitat profile, current strength and reef rugosity (3-D habitat complexity) do not show an east–west pattern (Jones, Berkelmans & Houston, 2011) that explains the genetic differences observed between inner and outer Keppel Island populations, further research is required to address the possibility that maladapted genotypes are unable to survive despite cross-shelf dispersal and recruitment.

Mechanisms of recovery

A. millepora populations in the Keppel Island archipelago are genetically isolated from most other populations on the GBR (Fig. 1) and are therefore largely self-sustaining. Along the GBR, south easterly trade winds dominate throughout the year but are seasonally displaced by northerly monsoonal winds during the austral summer (Pickard, 1977). The nearest mid-shelf reefs to the Keppel Islands are those of the Capricorn Bunker Group, >65 km to the east. South easterly winds could theoretically drive recruitment between the Capricorn Bunkers and the Keppel Island group, but A. millepora is relatively rare in the former (M van Oppen, pers. obs., 2012) and these reefs therefore unlikely serve as a source of larvae for the Keppel Island populations. A. millepora has a relatively high dispersal potential due to its broadcast-spawning mode of reproduction and long larval competency period. We hypothesise that, in the Keppel Islands, other coral species with similarly high dispersal potential to A. millepora, as well as species that disperse over shorter spatial distances, will also consist of primarily self-sustaining populations (although the Capricorn Bunkers may be a source for high dispersal coral species that occur at higher abundance there). This suggests that the archipelago is vulnerable to perturbations that cause widespread high coral mortality, as recovery through the arrival of recruits from reefs outside the Keppel Islands will be slow.

The 2002 Nth Keppel Island sample (collected prior to the 2002 bleaching event) exhibited no evidence of a genetic bottleneck, which was unexpected given the high mortality experienced during the 1998 mass bleaching event (Berkelmans et al., 2004). The same population showed an FST value not significantly different from zero when compared with the 2008 sample from the same location. In addition to the 2002 bleaching event, a mass bleaching episode occurred in the Keppel Islands in 2006, causing ∼40% loss in coral cover (Jones, Berkelmans & Houston, 2011). Given that severe bleaching reduces reproductive output in the subsequent spawning season (Michalek-Wagner & Willis, 2001; Jones & Berkelmans, 2011), and that it would take at least 2–3 years for new recruits to reach reproductive maturity even for the fast-growing Keppel Island Acropora spp. (Omori, 2010), there was little scope for local colonies that survived the 2002 and 2006 bleaching events to contribute to coral recovery through larval recruitment by 2008. This, in combination with the lack of evidence for recent genetic bottlenecks in all Keppel Island populations studied here (which were collected in 2008 and 2009), supports the hypothesis that in spite of reports of widespread mortality, whole colony mortality was actually low following the 2002 and 2006 bleaching events (although visual surveys that did not examine cryptic remnant tissues indicated whole colony mortality was high) and that tissue regrowth, rather than external recruitment, was the main mechanism of recovery following the two bleaching events. This supports the work of Diaz-Pulido et al. (2009) showing that coral recovery had occurred unexpectedly rapidly (within 12 months) after bleaching from surviving tissues in apparently dead colonies. Coral recruitment during this period was low (Diaz-Pulido et al., 2009) and instead, recovery must have occurred through regrowth from cryptic remnant tissues, as supported by our genetic data. The unusually high growth rates of Acropora spp. in the Keppel Islands (Diaz-Pulido et al., 2009; Jones & Berkelmans, 2010) appear to be key to this atypically rapid coral cover recovery following disturbance.

Preliminary observations show that the speed of recovery following flood events is slower than that following bleaching, likely reflecting the more common occurrence of whole colony mortality in areas affected by fresh water inundation, despite its more spatially restricted impact. For example, the 1991 flooding event (Byron & O’Neill, 1992; Furnas, 2003; Jones & Berkelmans, 2014) caused total mortality of Acropora spp. down to 1.3 m below lowest tide (van Woesik, DeVantier & Glazebrook, 1995). Average coral cover at the southern/western side of Nth Keppel Island (site 4 in Byron & O’Neill, 1992) dropped from pre-flood levels of 51–75% to 10% post-flood (Byron & O’Neill, 1992) and had not yet fully recovered by February 1995 (∼40%, R Berkelmans, 1995, unpublished data). Similarly coral cover on the southern/western side of Halfway Island (site 20 in Byron & O’Neill, 1992) dropped from 76–100% before the 1991 flood to 50% post-flood but were fully recovered by August 1996 (∼84%, R Berkelmans, 1996, unpublished data). However, reefs on the northern and eastern sides of these islands generally showed little coral loss (Byron & O’Neill, 1992; van Woesik, DeVantier & Glazebrook, 1995). Our interpretation of these observations, in light of the population genetic results presented here, is that while whole colony mortality is more prominent during floods than bleaching, flooding has a spatially more variable impact within the Keppel Islands. The slower recovery of flood impacted southern and western sides of the islands was likely mostly due to larval recruitment from northern and eastern sites.

Management implications

The lack of evidence for genetic erosion in this study demonstrates that, despite four high mortality events including flooding in 1991, and bleaching in 1998, 2002 and 2006, the resilience of coral populations in the Keppel Islands was high prior to late 2008—early 2009 when the sampling for this study was conducted. However, in this isolated reef system, recruitment from external sources is limited, potentially placing future recovery at risk if disturbance events are too frequent or are severe enough to cause widespread whole-colony mortality.

The isolation of the Keppel Island archipelago and genetic distinctiveness of its coral populations have implications for reef restoration actions and management interventions that may be considered in the future. For instance, the introduction of coral genotypes from elsewhere, with the intent to accelerate recovery and boost resilience (Hoegh-Guldberg et al., 2008; van Oppen et al., 2014), may have positive effects as a consequence of introducing new gene variants into the Keppel Island populations if introduced colonies interbreed with the remaining native corals, but could also have adverse effects due to outbreeding depression. This requires testing under controlled conditions before such measures would be implemented. The Keppel Island corals possess a set of valuable traits, including genetic distinctiveness, high growth rates and recovery potential, which, in combination with their relative isolation from other reefs should afford these ecosystems a high conservation status.

Conclusions

Our microsatellite genotyping results demonstrate that populations of the common reef builder, A. millepora, in the Keppel Islands fall into two clusters with limited gene flow; those at the inner islands vs. those at the outer islands (i.e., Barren Island, Outer Rocks and Man & Wife Rocks). Further, populations of this species in the Keppel Island archipelago are self-sustaining and receive very little input from populations elsewhere on the GBR. Genetic diversity analyses suggest coral recovery in the Keppel Islands often occurs from surviving colony regrowth rather than by recruitment from external sources, especially following bleaching. However, when whole colony mortality is widespread within a reef but variable among reefs (as is the case with floods), recruitment from external, nearby reefs that suffered low mortality can facilitate recovery.

Supplemental Information

Table S1 The four triplex reactions, fluorescent labels, expected fragment sizes and repeat unit size for the twelve microsatellite loci used to identify genetic population structure of Acropora millepora.

Click here for additional data file.

Table S2 Basic statistics for all loci in the nine Keppel Island populations for which new data were obtained in this study

N, number of samples per locus and population after removal of repeated MLGs; A, number of alleles; HE, expected heterozygosity; HO observed heterozygosity; FIS, inbreeding coefficient. Italic font indicates statistical significance after FDR correction.

Click here for additional data file.

Table S3 Results of Linkage Disequilibrium tests for all loci in the nine Keppel Island populations for which new data were obtained in this study

Adjusted alpha was 0.011; significant values are highlighted in yellow.

Click here for additional data file.

Table S4 Pairwise FST values below diagonal, p-values above diagonal

Most values are statistically significant; non-significant values have shaded background, and p-values larger than adjusted α are printed in bold face (adjusted α = 0.012).

Click here for additional data file.

Table S5 Assignment probabilities of putative recent migrants as identified in GeneClass2

Assignment probabilities >0.1 are highlighted.

Click here for additional data file.

Figure S1 Genetic diversity in A. millepora populations across the GBR: dierence between mean allelic evenness and allelic evenness at each site (A), and mean allelic evenness at each site (B).

Click here for additional data file.

Figure S2 PCoA of pairwise D est values of the Keppel Island populations.

Click here for additional data file.

Figure S3 Isolation by distance analysis (IBD).

Click here for additional data file.

Figure S4 STRUCTURE LnProb (K) and ΔK and TESS DIC for Keppel Islands only.

Click here for additional data file.

Figure S5 STRUCTURE (A & C) and TESS (B & D) results of Keppel Island populations for K = 2 (A & B) and K = 3 (C & D).

Click here for additional data file.

Figure S6 STRUCTURE LnProb (K) and ΔK and TESS DIC plot for combined data.

Click here for additional data file.

Data S1 Raw microsatellite data for the combined data set (this study and van Oppen et al., 2011) in GenAlex format.

Click here for additional data file.

The authors would like to thank Scott Gardner for assistance with sampling.

Additional Information and Declarations

Competing Interests

Author Contributions

Field Study Permissions

The authors declare there are no competing interests.

Madeleine J.H. van Oppen conceived and designed the experiments, analyzed the data, contributed reagents/materials/analysis tools, wrote the paper, prepared figures and/or tables, reviewed drafts of the paper.

Vimoksalehi Lukoschek analyzed the data, wrote the paper, prepared figures and/or tables, reviewed drafts of the paper.

Ray Berkelmans conceived and designed the experiments, contributed reagents/materials/analysis tools, wrote the paper, reviewed drafts of the paper.

Lesa M. Peplow performed the experiments.

Alison M. Jones conceived and designed the experiments, performed the experiments, contributed reagents/materials/analysis tools, wrote the paper, reviewed drafts of the paper.

The following information was supplied relating to field study approvals (i.e., approving body and any reference numbers):

Great Barrier Reef Marine Park Authority collection permit numbers G09/30237.1 and G08/26114.1

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
