# Peer review of "A population genetic assessment of coral recovery on highly disturbed reefs of the Keppel Island archipelago in the southern Great Barrier Reef"

_PeerJ, doi:10.7717/peerj.1092_

## Round 0.1 · original submission · Minor Revisions

Congratulations on a very nice paper. Please address the minor suggestions by Reviewer 2 that will help you improve your manuscript.

·

Basic reporting

No particular comments. Manuscript was a welcome, easy read - clear reporting of goals and background.

Experimental design

The question about this population (regional reef system) of Acropora millepora is how much of the recovery is from outside of the system - recruitment - versus damaged colonies recovering and growing. So they will look at diversity and connectivity using microsatellites in 9 shallow areas of Keppel Islands as well as numerous other sites throughout GBR. The datasets were sufficiently large - 370 individuals in the Keppels, over 1300 total. Their analytical approach is mostly population genetic boilerplate, with the exception of the implementation of TESS as an alternative to STRUCTURE; this is explained well and a useful contrast in this system (though the results are largely concordant).

Validity of the findings

Though the concern over linkage disequilibrium and bottleneck dynamics are probably overkill with only 10 loci, it is worth evaluating any impact of the 4 major mortality events referenced for this reef. I'm a little bit skeptical of what demographic inference could be made from these data, but in the end the null hypothesis is not rejected so it doesn't really matter - and more importantly, the data also tend to support the idea that the Keppel population(s) are to date self-sustained through recovery growth, as they are somewhat isolated from the rest of the GBR.

The presentation of STRUCTURE/TESS results was solid and used a fairly standard implementation, which is good - the questions are about the biology rather than methods. They present their results clearly and without dogmatic attachment to any particular K value! The discussion of recovery dynamics is solid and interesting. Lines 346-355 were particularly salient as management-based introduction of diversity is pondered; I agree that such effort should only be done after considerable caution.

Additional comments

Though the paper is a relatively straightforward documentation of genetic diversity and connectivity for A. millepora across the GBR, I had to note that I really liked Figure 1 - very clear with the results and sampling presented side by side. Obviously Figure 2 is gorgeous as well but I'll credit the reef more in that case.

·

Basic reporting

This is an interesting and well-written article that uses genetic information to highlight the importance of coral reef conservation around the archipelago of the Keppel Islands (KI). The article is scientifically sound and sampling is adequate, especially when couple with a previous study of the Great Barrier Reef. I agree with most authors’ interpretations, but suggest the difference is not between KI and all other populations but between the western most side (Outer, Man & Wife and Barren) of the KI archipelago and all other populations, including all eastern KI and the GBR. The deepest break (Fig. 1) is between Corals from the eastern, including non-KI populations such High Peak and the western KI. Similarly, interpretations about rainfall are only appropriate about the eastern KI (most citations are for these islands) and not the western-outer islands. Authors should investigate on the environmental differences between eastern (inner) and western (outer) islands to better explain the genetic patterns.

Experimental design

Central to the conclusion of the article is the movement of individuals from or to the Keppel Islands, which is mostly based on interpreting STRUCTURE plots. I suggest authors should use a direct test for migration from/to eastern and western islands. Perhaps the program Bayesass (Wilson and Rannala 2003) designed to test for recent migration events can capture migration patterns across corals in the KI archipelago.

The authors suggest that KI may have lost genetic diversity as a result of environmental perturbations. The authors test for differences in allelic diversity from KI and the rest of the GBR and conclude that there is more genetic diversity in the GBR. The lower genetic diversity in KI can also be interpreted as a sampling artifact. The sampling effort in the GBR is substantially larger both in terms of number of individuals and also in geographical area (perhaps more habitats too). A more even test would be to compare High Peak and each of the KI populations or a test that compares segments of the GBR equal in geography and number of samples with that of the KI. Alternatively, I suggest testing for the presence of private alleles in KI as it would be more powerful to convey the idea of special management for the islands.

References

Wilson and B. Rannala 2003. Bayesian inference of recent migration rates using multilocus genotypes. Genetics 163: 1177-1191.

Validity of the findings

To validate their findings from STRUCTURE, authors should test for a panmictic population (K =1). I suggest to asses K=1 to K = n and provide the likelihood scores and the delta K graphs.

The authors used the program MegaBACE for the present study and the program CEQ8800 system software (version 10) for the previous one, which may have resulted on an adjustment of the allele sizes. To fully replicate this study, the authors should publish their entire dataset including the GBR populations as part of SI of this article.




Other comments:

Line 121-124. Authors used two different programs to score alleles from the KI and GBR. To ensure replication of the study, I suggest publishing the entire dataset as part of the present article.

Line 137. Table S2 has info about heterozygosity but I cannot locate the HWE info.

Line 143. Is there a reason for 22 individuals? If so please explain why.

Line 165. I observed that FST values in Table S4 may still be significant after Bonferroni correction. If so add such a comment to the manuscript and say FDR though less conservative is more biologically reasonable.

Line 180-182. Authors need to test for K =1 and provide likelihood scores along with delta K for each analysis.

Line 201. Is it two or three times? Two at … and one at … Unclear.

Line 207 -208. This is an uneven test. Just by chance you have more alleles in a larger area or by sampling more individuals.

Line 219 -223. There seems to be an IBD pattern from Fig. S2. I suggest testing for the significance of the r2 coefficient and base your interpretation on the result.

Line 264 – 265. The highest LD in pairwise-locus comparisons is Rocks, opposite of what you are saying here. The interpretation of the LD in such a large (> 100) number of comparisons is weak. I suggest deleting this part.

Line 266-267. This is the most salient pattern of the study and it is even shared with fish. Looking for what differentiate western islands and eastern islands may help explain the pattern of genetic variation.

Line 277-280. I see this alternative entirely possible and my own work on reef cnidarias have showed it occurs at smaller (< 200 m) spatial scales (Prada and Hellberg 2014)

Line 336. Authors should consider environmental differentiation in the two parts of the KI archipelago. With eastern sites more prone to flood than western ones. Maybe other environmental features apply. Most of their info (citations) about flood comes from the eastern side. Knowing whether there are floods on the western side is important to understand genetic variation across the KI.

Line 350. Affects or effects?

Line 352. Efforts should be to preserve the unique allelic diversity of the KI not to bring other probably maladapted genotypes. It seems feasible for larvae from 21-121 or high peak to get to the western KI but larvae do not stay.

Line 361. Differentiation is not across the whole KI but only for the most western islands. I suggest testing for private alleles across the archipelago may provide stronger evidence to suggest higher conservation priority.



References

Prada, C. and M. E. Hellberg. 2014. Strong natural selection on juveniles maintains a narrow adult hybrid zone in a marine broadcast spawner. American Naturalist. 184: 702-713.

---

## Round 0.2 · accepted · Accept

I congratulate you on a very valuable contribution.